# Uptake of Human Papilloma Virus vaccine among young women living in fishing communities in Wakiso and Mukono districts, Uganda

**Muteebwa Laban**[1]*, **Gertrude Nanyonjo**[2], **Mathias Wambuzi**[2], **Ali Ssetaala**[2], **Geofrey Basalirwa**[2], **Dan Muramuzi**[3], **Jacqueline Kyosiimire Lugemwa**[4], **Brenda Okech**[2], **Ali Mirzazadeh**[5]

1 School of Medicine, College of Health Sciences, Makerere University, Kampala, Uganda, 2 Department of Community Studies, UVRI-IAVI HIV Vaccine Program, Entebbe, Uganda, 3 School of Public Health, College of Health Sciences, Makerere University, Kampala, Uganda, 4 Department of In-vitro studies, Uganda Virus Research Institute, Entebbe, Uganda, 5 Department of Epidemiology and Biostatistics, Institute for Global Health Sciences, University of California San Francisco, San Francisco, California, United States of America

* mutlabans@gmail.com

## Abstract

Human Papilloma Virus (HPV) is a preventable cause of cervical cancer, the commonest cancer among women in Uganda. The Uganda Ministry of Health included the HPV vaccine in the free routine immunization schedule since 2015. Five years after this policy, we assessed the uptake of the HPV vaccine and associated socio-demographic factors among young women living in fishing communities in Central Uganda in 2020. We analyzed secondary data from 94 young women aged 9–25 years who were recruited from the two fishing communities (Kasenyi landing site and Koome Island) in a primary study that aimed to promote awareness of maternal and childhood vaccines. We assessed uptake of the HPV vaccine as the proportion of participants who self-reported to have ever received at least one dose of the HPV vaccine. We assessed the socio-demographic factors associated with HPV vaccine uptake using a modified Poisson regression model adjusted for clustering by study site in STATA version 17. The mean (standard deviation) age of study participants was 21.1 (3.1) years and most (81.9%) of them were from Kasenyi landing site. The uptake of the HPV vaccine was 10.6% [95% Confidence Interval (CI) 5.6, 18.9]. After adjusting for covariates, being 13–19 years old (adjusted prevalence ratio [aPR] 5.52, 95%CI 1.69, 18.00) and of Catholic religion (aPR 5.55, 95%CI 1.53, 20.16) were significantly associated with HPV vaccine uptake. The HPV vaccine uptake was very low, despite the reported 99% national coverage of HPV vaccination program for the first dose at the end of 2019. Age and religion showed to be important determinants of the HPV vaccine uptake. Reasons for such very low uptake of HPV vaccinations need to be carefully assessed to find effective strategies to improve it.

**Data Availability Statement:** The data underlying the findings of this study can be accessed from a repository using DOI: 10.5061/dryad.34tmpg4rq.

**Funding:** The study was funded by Immunizing Pregnant Women and Infants Networks (IMPRINT) grant No.S22680. We also appreciate the support from the International AIDS Vaccine Initiative (IAVI) and the University of California, San Francisco's International Traineeship in AIDS Prevention Studies (ITAPS) (grant No U.S. NIMH, R25 MH0123256). The funders had no role in study design, data collection and analysis, decision to publish or preparation of the Manuscript. The content is solely the responsibility of the authors and does not necessarily represent the official views of the National Institutes of Health.

**Competing interests:** The authors have declared that no competing interests exist.

## Background

Human Papilloma Virus (HPV) is the cause of almost all cervical cancers and is responsible for an important fraction of other anal-genital, head, and neck cancers [1]. Cervical cancer is the fourth most common cancer among women worldwide with an age-standardized incidence rate of 13.3 per 100,000 [2]. HPV subtype16 and 18 contribute to over 70% of all cervical cancer cases worldwide [3] and are vaccine-preventable. East Africa has the highest age-standardized incidence rate of cervical cancer in Africa (40.1 per 100,000) [2]. In Uganda, cervical cancer is the most common cancer among females aged 15–44 years, with an age-standardized incidence rate of cervical cancer being 56.7 per 100,000 and an age-standardized mortality rate is 41.4 per 100,000 [1].

Other risk factors for cervical cancer include cigarette smoking, early sexual debut, high fertility rate, and infection with Human Immunodeficiency Virus (HIV) [4]. HIV-infected women have a four to five times greater risk of developing cervical cancer compared to uninfected women [5]. Young women living in fishing communities in Uganda have higher incidence of HIV compared to others in the general population and thus HPV vaccination in this HIV high-risk population could reduce the future burden of HIV and cervical cancer co-infection [5]. Females in fishing communities in Uganda have an early sexual debut compared to their male age-mates [6]. In addition, young women in fishing communities have high levels of mobility, which limits access to health services [7]. Despite these vulnerabilities that increase the risk of acquiring HIV and HPV infections, the uptake of HPV vaccine has not been documented among young women living in fishing communities.

HPV vaccination is the primary prevention strategy for cervical cancer. The World Health Organization (WHO) recommends HPV vaccination for girls aged 9–14 years as the most cost-effective public health intervention against cervical cancer in low-income countries like Uganda [3]. HPV vaccination in Uganda was initiated in 2008 through a demonstration project that led to the adoption of two vaccine delivery strategies, that is, school grade (grade 4) and age-based, for scale-up by the Ministry of Health (MoH) [8]. MoH recommends two doses of the HPV vaccine and has integrated into the routine immunization program since 2015 [9]. By end of 2019, the MoH estimated the coverage of the HPV vaccination program for first dose to be 99% [10]. However, recent community-based surveys conducted in Uganda have found low levels of HPV vaccine uptake [11,12]. Additionally, the hierarchical modeling of 2016 Uganda Demographic Health Survey (UDHS) data involving 16,093 adolescent girls 10–14 years old nested in 689 communities found that 78% of them were not vaccinated against HPV infection [13]. We sought to assess the level of uptake of the HPV vaccine among young women living in fishing communities and associated socio-demographic factors.

## Methods

### Study design and setting

We analyzed secondary data from a cross-sectional community-based implementation study conducted between January and February 2020. The study aimed at promoting awareness of maternal and childhood vaccines in fishing communities along Lake Victoria in Central Uganda. The participants were recruited from the Kasenyi landing site in Wakiso district and Koome Island in Mukono district. Kasenyi landing site is located in Wakiso district, about 33 km from Kampala city, and has a population of about 26,575 people. Koome Island has located in Mukono district 51.8 km from Kampala city and has a population of 2148 people. The main study enrolled both men and women aged 9–49 years who had stayed in the fishing community for at least one month before the start of data collection.

## Study population, inclusion, and exclusion

Data of young women aged 9–25 years who had a response on HPV vaccination status recorded were analyzed. This age bracket was chosen considering the period since the introduction of the HPV vaccine in Uganda in 2008. Out of the 134 young women recruited in the main study, only 94 participants had data for the HPV vaccination status recorded and were included in the analysis. Participants with missing data on HPV vaccination status were excluded from the study.

## Sampling and data collection

Data of all participants who fulfilled the eligibility criteria were extracted from the open data kit (ODK) that included the HPV vaccination status variable and the socio-demographic characteristics. All participants that fulfilled the eligibility criteria were included in the analysis (census sampling).

## Study variables

The dependent variable was the uptake of the HPV vaccine among young women which was defined as the proportion of participants who self-reported to have ever received at least one dose of the HPV vaccine.

Independent variables included age, religion, education level, the community of residence, period of stay in the community, tribe, occupation, having a child, ever heard about immunization of girls against HPV, ever heard about cervical cancer screening, ever having been screened for cervical cancer.

## Data analysis

Data were analyzed in STATA version 17.0 (Texas, USA). The data were declared as survey data, numeric variables were summarized using mean (standard deviation (SD)), categorical variables were summarized as frequencies and percentages adjusted for clustering by study site. At bivariate analysis, the association between the uptake of the HPV vaccine and the socio-demographic characteristics was assessed using modified Poisson regression using linearized standard errors to estimate the prevalence ratios (PR). The crude prevalence ratios (cPR) and their corresponding 95% CI were estimated. The independent variables with a P value of <0.2 were considered for the multivariable analysis. At multivariate analysis, the interaction was assessed using a manual stepwise elimination method. Independent variable that changed the prevalence ratio of any other variable already in the model by a magnitude of >10% was retained in the final model. Adjusted prevalence ratios (aPR) and their corresponding 95% CIs were reported. Variables with a P value of <0.05 was considered to be statistically significant in the multivariate model. Sensitivity analysis was done and participants who were excluded due to missing outcome status were not significantly different from those that were analyzed.

## Ethical considerations

The study protocol was approved by the Uganda National Council of Science and Technology (SS#5198) and the Uganda Virus Research Institute Research Ethics Committee (UVRI-REC#753). Participants were given the study information to help them make an informed decision and they were informed that participation was voluntary. Participants aged 18 years and above provided written informed consent to participate in the study. Participants below 18 years provided written informed assent in addition to the written informed consent

provided by their guardians or parents. The data used in the analysis were extracted from the ODK files by the data manager and these didn't include participant identifiers.

## Results

### Socio-demographic characteristics of study participants

The mean (SD) age of study participants was 21.1 (3.1). Most of the respondents were 20–25 years old (74.5% (70/94)), from the Kasenyi landing site (81.9% (77/94)) and 64.9% (61/94) had stayed in the study communities for <5 years. Most participants had less than secondary level education (59.6% (56/94)) and were engaged in businesses (71.3% (67/94)) other than the fishery business. Most participants had at least one child (74.5% (70/94) and had ever heard about the immunization of girls against HPV (64.9% (61/94)) (Table 1).

**Table 1. Characteristics of young women (N = 94) who participated in a survey conducted in January-February 2020 in Kasenyi Landing and Koome Island.**

| Variables | Frequency (N = 94) | Percentage (%) |
|---|---|---|
| **Age** | | |
| 9–19 Years | 24 | 25.5 |
| 20–25 Years | 70 | 74.5 |
| **Period of stay in study community** | | |
| Below 5years | 61 | 64.9 |
| 5years and Above | 33 | 35.1 |
| **Community** | | |
| Landing site (Kasenyi) | 77 | 81.9 |
| Island (Koome) | 17 | 18.1 |
| **Religion** | | |
| Catholic | 33 | 35.1 |
| Anglican | 22 | 23.4 |
| Other Religions* | 39 | 41.5 |
| **Tribe** | | |
| Baganda | 37 | 60.6 |
| Other tribes** | 57 | 39.4 |
| **Occupation** | | |
| Fishery business | 27 | 28.7 |
| Other businesses*** | 67 | 71.3 |
| **Education level** | | |
| <Secondary | 56 | 59.6 |
| ≥Secondary | 38 | 40.4 |
| **Ever heard about Immunization of girls against HPV** | | |
| Yes | 61 | 64.9 |
| No | 33 | 35.1 |
| **Has at least one child** | | |
| Yes | 70 | 74.5 |
| No | 24 | 25.5 |

*Moslem, Seventh Day Adventist, Born Again, Traditional.

**Munyankole, Musoga, Mukiga, Munyoro, Mugwere, Iteso, Lugbara.

***Bar attendant, Farmer, saloon attendant.

### HPV vaccine uptake

The uptake of the HPV vaccine was 10.6% (10/94), 95% CI (5.6, 18.9), of which seven out of ten had received only one of the required two doses, and almost all participants were from the Kasenyi landing site (9/10).

### Factors associated with HPV vaccine uptake

At multivariate analysis being 13–19 years old [aPR 5.52, 95%CI 1.69, 18.00] and of Catholic religion [aPR 5.55, 95%CI 1.53, 20.16] were significantly associated with uptake of HPV vaccine (Table 2).

## Discussion

The uptake of HPV vaccine among young women living in fishing communities was very low (only one in ten women had received at least one dose) compared to the Ministry of Health target of eight in ten women receiving two doses [9]. The uptake was even lower among women who were 20–25 years old. Individuals affiliated with the Catholic religion had higher HPV vaccine uptake compared to other religions. The low HPV vaccine uptake in this study is consistent with other community-based surveys conducted in Eastern (14%) and Northern (17.6%) Uganda [11,12]. However, the uptake in this study was even lower than that reported by Scott and colleagues in a cross-sectional study conducted in Uganda (88.9%) and other lower and middle-income countries (77.2% in India to 96.1% in Viet Nam) [14]. The high uptake of the HPV vaccine reported in these countries could have been because the survey was conducted immediately after an HPV vaccine demonstration project involving mass immunization of young girls in school and communities through campaign programs which was not the case for this current survey. Additionally, the uptake of the HPV vaccine in this current study was lower than that reported by a study conducted in Central Uganda (43.3%) [15]. The difference in the findings could be due to the fact that this study was conducted in an adolescent clinic setting whereas our study was a community-based survey. The low uptake of HPV vaccines among young women may also be attributed to limited awareness of the services evidenced by the fact that 1 in 3 young women reported not to have ever heard about HPV vaccines. None the less women living in fishing communities along the Lake Victoria basin are characterized by high levels of mobility (outmigration) which limits access to health services [16,17]. A qualitative study exploring barriers to HPV vaccine uptake in Northern Uganda reported individual level factors like inadequate knowledge about the vaccine, frequent mobility between vaccine doses, school absenteeism (where school-based approach is used) and drop-out, fear of injection pain and discouragement from caregivers or peers [18].

Conversely, the uptake of the HPV vaccine in this study is higher than the pooled HPV vaccine uptake reported in a recent systematic review of peer-reviewed studies published up to October 2020 in low-income countries (3.48%, 95% CI: 1.16, 5.66) [19]. However, this systematic review reported a high heterogeneity and high risk of bias among the published studies considered, and this may have affected the pooled estimate.

Young women aged 20–25 years were less likely to be vaccinated against HPV compared to adolescent girls 13–19 years. This could be because the HPV vaccine campaigns in Uganda targeted adolescents in lower primary (school-based strategy) and thus these young adults could have missed vaccination. Furthermore, the school attendance of adolescents in lower classes (lower primary) is more than in higher classes thus more likely to be vaccinated compared to the young adults aged 20–25 years since vaccination campaigns targeted lower primary level.

**Table 2. Bivariate and Multivariate analysis of socio-demographic factors associated with the uptake of HPV Vaccine among young women in Kasenyi Landing and Koome Island.**

| Variable | Bivariate analysis | | Multivariate analysis | |
|---|---|---|---|---|
| | cPR(95% CI) | P Value | aPR (95%CI) | P Value |
| **Age** | | | | |
| 9–19 Years | 2.92 (0.90, 9.45) | 0.074 | 5.52 (1.69, 18.00) | **0.005** |
| 20–25 Years | Ref | | Ref | |
| **Period of stay in study community** | | | | |
| <5year | 2.16 (0.47, 9.88) | 0.315 | | |
| 5 years and above | Ref | | | |
| **Community** | | | | |
| Landing site | 1.99 (0.25, 15.94) | 0.514 | | |
| Island | Ref | | | |
| **Religion** | | | | |
| Other religions* | Ref | | Ref | |
| Catholic | 2.96 (0.59, 14.7) | 0.183 | 5.56 (1.53, 20.16) | **0.010** |
| Anglican | 2.66 (0.47,15.20) | 0.268 | 2.49 (0.45, 13.72) | 0.290 |
| **Tribe** | | | | |
| Other tribes** | Ref | | | |
| Baganda | 1.54 (0.47, 5.07) | 0.473 | | |
| **Occupation** | | | | |
| Other businesses | Ref | | | |
| Fishery business | 1.06 (0.29, 3.90) | 0.925 | | |
| **Education level** | | | | |
| <Secondary | Ref | | Ref | |
| ≥Secondary | 2.21 (0.65, 7.48) | 0.199 | 3.05 (0.85,10.95) | 0.087 |
| **Ever heard about Immunization of girls and women against HPV** | | | | |
| Yes | 4.87 (0.62, 8.26) | 0.131 | | |
| No | Ref | | | |
| **Have at least one child** | | | | |
| Yes | Ref | | | |
| No | 1.94 (0.58, 6.48) | 0.275 | | |
| **Ever had Cervical cancer screened*** | | | | |
| Yes | 1.17 (0.31, 4.34) | 0.816 | | |
| No | Ref | | | |
| **Ever heard about cervical cancer screenings among women** | | | | |
| Yes | 1.58 (0.21, 1.94) | 0.657 | | |
| No | Ref | | | |

cPR-Crude Prevalence Rati; aPR-Adjusted Prevalence Ratio; Ref-Referent group

*Moslem, Seventh Day Adventist, Born Again, Traditional

**Munyankole, Musoga, Mukiga, Munyoro, Mugwere, Iteso, Lugbara.

Non-Catholics were less likely to be vaccinated compared to other religions. A qualitative study conducted in Northern Uganda reported that negative religious beliefs were some of the community-level barriers to the uptake of HPV vaccine [18]. In Uganda, 37.1% of the women aged 15–49 years are affiliated with the Catholic religion while only 13.7% are Muslims [20]. A recent study among sub-Saharan African countries reported that Christians were more likely to uptake the HPV vaccine compared to the Muslims [21]. This further emphasizes the importance of religious influence on the uptake of health interventions including vaccines.

## Limitations

The sample size was very small and this could have increased random error and smaller effects could not be assessed. There could have been information bias due to the self-reported nature and a long period of recall of the outcome and missing data (29.8% missing data on the uptake of HPV vaccine). However, sensitivity analysis was done and the characteristics of participants (age, religion, education level, the community of residence, period of stay in the community, tribe, occupation, having a child) who had missing data were not statistically different from those that were analyzed. Some potential confounder variables that could have influenced uptake were not measured and so not controlled for and thus the reported estimates may be biased for example health system and other individual factors like socio-economic status. However, the biases did not significantly influence the study findings substantially.

## Conclusion

The uptake of the HPV vaccine was very low despite inclusion of the HPV vaccine in the routine immunization schedule and the reported 99% national coverage of HPV vaccination program for the first dose at the end of 2019. This threatens the ambitious goal set by the WHO of reducing the incidence of cervical cancer to less than 4 per 100000 women years by 2030. The involvement of religious leaders may be essential in increasing the uptake of the HPV vaccine among eligible girls.

We recommend that the Ministry of Health and implementing partners prioritize vaccination of HIV high-risk young women like those living in fishing communities after assessing and addressing the barriers to access and uptake of HPV vaccine. A community engagement approach that involves religious leaders should be considered to help dispel myths and encourage vaccination against HPV. Furthermore, The Ministry of Health should consider interventions like intensifying health education campaigns and outreaches, improving access to vaccination services and where possible include incentives to improve HPV vaccination among this high risk sub-population.

## Acknowledgments

The authors are very grateful to the IMPRINT network and management of Uganda Virus Research Institute-International AIDS Vaccine Initiative (UVRI-IAVI) HIV Vaccine Program, for supporting the implementation of the primary study and allowing access to the data.

We are indebted to Gertrude Nanyonjo who was the Principal Investigator of the study for conceptualizing the idea and the leadership provided during the implementation of the project. We thank all the stakeholders at community level in landing sites and at the Island that we worked with for their support, the study participants for providing the data; the Research and Ethics Committees of UVRI and the Community Advisory Board (CAB) for their continuous advice and guidance throughout the implementation of the study.

The contents of this manuscript are the responsibility of the authors and do not necessarily reflect the views of the IMPRINT network. We thank the reviewers for their insightful comments on the manuscript. We are greatly indebted to the field research team for their tireless efforts in collecting data. This work was funded by IMPRINT network (https://www.imprint-network.co.uk).

## Author Contributions

**Conceptualization:** Gertrude Nanyonjo, Mathias Wambuzi, Ali Ssetaala, Brenda Okech.

**Data curation:** Geofrey Basalirwa.

**Formal analysis:** Muteebwa Laban, Ali Ssetaala, Geofrey Basalirwa, Dan Muramuzi, Jacqueline Kyosiimire Lugemwa, Ali Mirzazadeh.

**Funding acquisition:** Gertrude Nanyonjo, Ali Ssetaala, Brenda Okech, Ali Mirzazadeh.

**Methodology:** Muteebwa Laban, Mathias Wambuzi, Geofrey Basalirwa, Dan Muramuzi, Jacqueline Kyosiimire Lugemwa, Brenda Okech, Ali Mirzazadeh.

**Project administration:** Gertrude Nanyonjo, Mathias Wambuzi, Ali Ssetaala, Brenda Okech.

**Supervision:** Gertrude Nanyonjo, Mathias Wambuzi, Ali Ssetaala, Brenda Okech, Ali Mirzazadeh.

**Validation:** Muteebwa Laban, Dan Muramuzi, Jacqueline Kyosiimire Lugemwa, Brenda Okech.

**Writing – original draft:** Muteebwa Laban, Dan Muramuzi, Jacqueline Kyosiimire Lugemwa, Ali Mirzazadeh.

**Writing – review & editing:** Muteebwa Laban, Gertrude Nanyonjo, Mathias Wambuzi, Ali Ssetaala, Geofrey Basalirwa, Dan Muramuzi, Jacqueline Kyosiimire Lugemwa, Brenda Okech, Ali Mirzazadeh.

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
