## [Decision Letter · Decision Letter 0]

3 Jan 2024

PGPH-D-23-02173

Uptake of Human Papilloma Virus vaccine among young women living in fishing communities in Wakiso and Mukono districts, Uganda

Dear Dr. Laban,

Thank you for submitting your manuscript to PLOS Global Public Health. After careful consideration, we feel that it has merit but does not fully meet PLOS Global Public Health’s publication criteria as it currently stands. Therefore, we invite you to submit a revised version of the manuscript that addresses the points raised during the review process.

We look forward to receiving your revised manuscript.

Kind regards,

Edward S. Peters, DMD, SM, ScD

Academic Editor

Journal Requirements:

Additional Editor Comments (if provided):

Reviewers' comments:

Reviewer's Responses to Questions

**Comments to the Author**

1. Does this manuscript meet PLOS Global Public Health’s publication criteria? Is the manuscript technically sound, and do the data support the conclusions? The manuscript must describe methodologically and ethically rigorous research with conclusions that are appropriately drawn based on the data presented.

Reviewer #1: Partly

Reviewer #2: Yes

2. Has the statistical analysis been performed appropriately and rigorously?

Reviewer #1: No

Reviewer #2: Yes

3. Have the authors made all data underlying the findings in their manuscript fully available (please refer to the Data Availability Statement at the start of the manuscript PDF file)?

Reviewer #1: No

Reviewer #2: Yes

4. Is the manuscript presented in an intelligible fashion and written in standard English?

Reviewer #1: Yes

Reviewer #2: Yes

5. Review Comments to the Author

Reviewer #1: This study investigated HPV vaccination uptake in fishing villages of Uganda in order to understand if uptake was as high as nationally reported; and factors associated with uptake. This data is important in understanding how vaccination uptake may vary across a country. There are some significant concerns with the analyses conducted and interpretations provided. I have detailed these and other comments below:

Introduction:

Please check the sentence on line 59-60. High mobility in young women limits access to health services? Is this correct?

Line 70: MoH recommends 2 doses of which HPV vaccine? This an important detail.

Methods:

The currently stated analyses is fine but I am wondering why the authors did not conduct a logistic regression to determine which factors were associated with uptake (did vs. did not get the vaccine). It is my understanding that they know who did and did not take the vaccine (via self-report), as state in the Line 92: inclusion of those “who had a response”. Would that not mean they also had those that reported no?

Results:

It would be beneficial to not the number of women who reported taking the vaccine, not just the percentage. This could be added to Table 2 or mentioned with the percentage on line 153.

The interpretation you provided starting at Line 157 appears to be an interpretation of logistic regression of the whole study population. Per Table 2, it appears you only modeled those that answered “yes” to vaccination—meaning your interpretation cannot indicate factors associated with uptake without a “control” or “negative” group. If I’m misunderstanding then you need to add the same data under “HPV uptake” for that that answered “no” either in Table 2 or Table 1.

Based on the reported N for “other businesses” in occupation I suggest creating one more category breaking this up.

The same comment for Religion, especially because in the Discussion you indicate Muslim religion tends to differ in uptake.

Table 2: The percentages in the “HPV uptake” column do not make sense. For example (age), 5 is not 20.8% of 10 (the total at the top).

Table 2: I suggest make the reference groups in the analyses so that all PRs are >1. This simplifies interpretation for the reader.

Discussion:

Line 172-175: I suggest removing or softening this statement. This is a strong claim considering your data is self-reported, from only 2 communities in Uganda, and was conducted at one time point. The WHO’s target is a global target and many other areas have much higher vaccinataion uptake.

Line 185-187: Could the difference simply represent a difference of uptake across regions in Uganda? There does not have to be a methdological reason...it could simply be fact that practices differ across Uganda.

Line 195-197: You mention age of vaccination in Uganda for the women you targeted. I wonder, Is the vaccine strategy still the same in Uganda now as when these women would've been targeted for vaccination? Perhaps your data suggest expansion of the ages targeted.

Limitations: Self-report is a large issue here and should be mentioned specifically.

Another is the fact that this data comes from only 2 communities in Uganda and is not representative of the whole country or even other communities.

Line 217-221. This is not clear—potential confounders were not meansured and so controlled for? Is this a typo? Also how do you know the biases did not influence study findings substantially--there was no bias analyses and many factors were not able to be accounted for.

Reviewer #2: Comments to the Author:

Thanks for giving the opportunity to review the manuscript. We suggest that the paper has great potential to be published in this journal .

The article addresses an important public health issue – the low uptake of the Human Papillomavirus (HPV) vaccine among young women in fishing communities in Uganda. The study provides relevant background information, methods, results, and discussions. Below are some specific points for consideration:

1. Title: Consider making the title more specific by including information about the fishing communities or emphasizing the focus on sociodemographic factors. For example, " Unraveling HPV Vaccine Uptake Challenges: Insights from Fishing Communities in Uganda” or “Low Uptake of HPV Vaccine in Fishing Communities: A Sociodemographic Analysis in Wakiso and Mukono Districts, Uganda."

2. Abstract: Clarify the term "primary implementation study" in the abstract. Provide a brief explanation or definition for readers who may not be familiar with the term.

3. Introduction: Provide a broader context of the HPV vaccination landscape in Uganda. Discuss any existing challenges, successes, or trends in HPV vaccine uptake nationally or in similar communities.

4. Methods: Consider breaking down the methods section into subsections (e.g., Study Design, Participants, Data Collection, Data Analysis) to enhance clarity and organization.

5. Results: Include tables or graphs to visually present key results, such as the sociodemographic characteristics of study participants, age distribution, and HPV vaccine uptake rates.

6. Discussion: Comparison with Other Studies: Compare the findings of this study with those of similar studies in Uganda or other regions to provide a broader perspective on HPV vaccine uptake.

7. Discuss potential reasons for the low HPV vaccine uptake observed in fishing communities, such as access barriers, awareness issues, or community-specific challenges.

8. Propose more detailed strategies for improving HPV vaccine uptake in fishing communities based on the study findings. This could include community engagement programs, targeted awareness campaigns, or collaboration with religious leaders.

9. Explicitly acknowledge potential biases in the limitations section and discuss how these biases may have influenced the study results.

10. Condense the recommendations section to emphasize key points. Clearly outline the proposed interventions for improving HPV vaccine uptake.

Overall Impression: The study addresses a significant public health concern and provides valuable insights into the factors influencing HPV vaccine uptake. Enhancements in presentation, clarity, and expansion of certain sections could further strengthen the article. Overall, the work contributes to the understanding of vaccination challenges in specific populations and highlights the need for targeted interventions.

6. PLOS authors have the option to publish the peer review history of their article (what does this mean?). If published, this will include your full peer review and any attached files.

**Do you want your identity to be public for this peer review?** For information about this choice, including consent withdrawal, please see our Privacy Policy.

Reviewer #1: No

Reviewer #2: **Yes: **Subash Chandra Sonkar

---

## [Editor Report · Decision Letter 1]

13 Feb 2024

PGPH-D-23-02173R1

Uptake of Human Papilloma Virus vaccine among young women living in fishing communities in Wakiso and Mukono districts, Uganda

Dear Dr. Laban,

Thank you for submitting your manuscript to PLOS Global Public Health. After careful consideration, we feel that it has merit but does not fully meet PLOS Global Public Health’s publication criteria as it currently stands. Therefore, we invite you to submit a revised version of the manuscript that addresses the points raised during the review process.

We look forward to receiving your revised manuscript.

Kind regards,

Edward S. Peters, DMD, SM, ScD

Academic Editor

Journal Requirements:

Additional Editor Comments (if provided):

Please revise table 2. It is unclear and there is no need for p-values an CI. Also, the response did not sufficiently address the prior reviews concerns. Please revise 

Reviewers' comments:

Please revise table 2. It is unclear and there is no need for p-values an CI. Also, the response did not sufficiently address the prior reviews concerns. Please revise 

---

## [Editor Report · Decision Letter 2]

21 Mar 2024

Uptake of Human Papilloma Virus vaccine among young women living in fishing communities in Wakiso and Mukono districts, Uganda

PGPH-D-23-02173R2

Dear Laban,

We are pleased to inform you that your manuscript 'Uptake of Human Papilloma Virus vaccine among young women living in fishing communities in Wakiso and Mukono districts, Uganda' has been provisionally accepted for publication in PLOS Global Public Health.

Best regards,

Edward S. Peters, DMD, SM, ScD

Academic Editor